# Foreign Direct Investment, Natural Resources, Economic Freedom, and Sea-Access: Evidence from the Commonwealth of Independent States

**Wencong Lu [1],[†], Ikboljon Kasimov [2],[*],[†] , Ibrokhim Karimov [3] and Yakhyobek Abdullaev [4]**

[1]  School of Public Affairs, Zhejiang University, Hangzhou 310058, China; wenclu@zju.edu.cn
[2]  School of Management, Zhejiang University, Hangzhou 310058, China
[3]  National School of Development, Peking University, Beijing 100871, China; ibrokhim2016@isscad.pku.edu.cn
[4]  School of Economics and Trade, Hunan University, Changsha 410000, China; yahyoabdullaev@gmail.com
[*]  Correspondence: 11720063@zju.edu.cn or ikboljon.kasimov@hotmail.com
[†]  Authors, Lu and Kasimov, have contributed equally to this study.

**Abstract:** This study examines the importance of natural resources, economic freedom, and sea-access in attracting foreign direct investment (FDI) inflows to the Commonwealth of Independent States (CIS), using panel data from 1998 to 2017. The Prais-Winsten regression with panel-corrected standard errors (PCSEs) is employed for all estimations. Feasible Generalized Least Squares (FGLS), Random Effects with Driscoll-Kraay standard errors (RE (D-K)), and Random Effects of Generalized Least Squares (RE (GLS)) estimators are used to test the sensitivity of PCSEs' estimates to changes in the underlying empirical model, whereas Instrumental Variables with Two Stage Least Squares (IV (2SLS)), Limited Information Maximum Likelihood (LIML), and Baltagi's Two-Stage Least-Squares Random-Effects (IV (EC2SLS)) estimators are used to address potential endogeneity concerns. The estimates confirm that natural resources, economic freedom, and sea-access are robust and decisive factors affecting FDI location decisions of foreign investors in CIS. More precisely, the results suggest that increased revealed comparative advantage in petroleum, higher economic freedom characterized by the increased government size and open markets, and territorial coastlines have a statistically significant and positive effect on FDI inflows to CIS transition economies. We also find that direct access to the Black Sea and the Caspian Sea provides a significant geographic competitive advantage to Azerbaijan, Kazakhstan, Georgia, Russia, Turkmenistan, and Ukraine in attracting FDI inflows over the other CIS member-states.

**Keywords:** foreign direct investment; natural resources; oil; economic freedom; sea-access; PCSE; CIS

## 1. Introduction

During the last three decades, FDI inflows have experienced an unprecedented upsurge and provided a vigorous impetus for domestic investments, capital accumulation, and sustainable economic growth in developing countries [1]. FDI is also perceived to facilitate modern production and management practices transfer, increase employment, improve labor standards, and provide vast prospects for ameliorating the state of the services sector, such as banking and finance, telecommunications, and transportation [2]. Given these concomitant benefits, various strands of theoretical and empirical literature have been developed around the determinants of FDI [3–5]. Still, empirical evidence on the significance of FDI determinants has not been conclusive. Besides, most existing studies focus on developed and developing countries and, despite the long history and rich literature on FDI, there is a dearth of econometric research on transition economies, particularly on CIS countries. Hence, our knowledge of both which factors affect inward FDI and how they

affect inward FDI to CIS member-states is equivocal. Considering that FDI is an essential source of financing in the transition process from centrally planned to the market-based economy and the achievement of sustainable development goals [6], how to effectively promote and realize FDI is worth the consideration of policymakers and scholars. Therefore, using a robust new methodology, this study examines the impact of natural resources, economic freedom, and sea-access on FDI inflows to CIS economies.

Empirically, the natural resources and FDI nexus has received considerable attention [7–10]; however, with the evidence of inducing positive and negative effects, there is no common consensus over the impact of natural resources on inward FDI. Undeniably, natural resources are critical components of the economy, especially in developing countries where the resource extractive sector constitutes a considerable portion of the gross domestic product (GDP) [6]. Nevertheless, some studies on FDI have found that resource-rich countries attract a lower volume of inward FDI than resource-scarce countries [11–13]. Considering that CIS countries are endowed with vast natural resource reserves [14], a substantial volume of FDI to these countries is mainly concentrated on resource extraction, signposting the importance of resources in attracting FDI [15]. Unfortunately, resource wealth does not automatically translate into national wealth; instead, resources tend to spread corruption, source conflicts and cause environmental damage [6,16]. Therefore, akin to many other resource-abundant countries, CIS member-states are often characterized by weak legal and regulatory structures that raise concerns over the investment climate quality [12]. To account for the quality of host countries' institutional environment, a growing strand of literature on FDI determinants [17–19] has employed economic freedom index and underlined its significance in attracting FDI. This index represents the degree to which a country is pursuing free-market principles, where a higher level of economic freedom implies the existence of a business environment conducive to prosperity [20]. Hence, on a priori grounds, economies with higher economic freedom index ought to be more enticing for multinational enterprises (MNEs) and facilitate a higher FDI inflows [21]. Howbeit, some studies have incorporated natural resources and institutions in different multivariable models; the importance of physical geographic location has been practically overlooked [22]. Geographic location characteristics, such as lack of territorial sea-access, could be a severe disincentive to socio-economic development [23]. Notably, the absence of coastlines increases the transport and other costs of doing business, isolates countries, and subsequently, undermines their ability to participate in global production networks [24,25]. Thus, despite the similarities in legal and governance structure, natural resource reserves, or macroeconomic performance, MNEs' choice of FDI location can be subject to the country-specific geographic location of host countries [26].

Given the prior literature, it is conspicuous that natural resources and economic freedom have grown in importance in studying the determinants of FDI inflows, and have particular relevance to the empirical studies of developing nations [13,21,27]. Nonetheless, thus far, econometric research linking natural resources, economic freedom, and physical geographic location to FDI is limited, particularly in transition economies. This research contributes to the existing literature by using a new and large panel dataset for 12 CIS countries from 1998 to 2017, integrating the role of natural resources, economic freedom, and sea-access in attracting FDI inflows, and applying a different empirical estimation model than that of prior studies. It addresses potential concerns over the robustness of regression results to changes in the underlying empirical model and accounts for possible endogeneity issues. Moreover, to the best of our knowledge, this is the first study using an incorporated analysis of natural resources, economic freedom, sea-access, and inward FDI in CIS, which provides a better understanding of FDI phenomenon in resource-rich CIS economies.

The remainder of the paper proceeds as follows. Section 2 provides a brief literature review on FDI determinants. Section 3 describes the data and their sources, explains the variables used in this study, and specifies the estimation model. The empirical results are discussed in Section 4, and Section 5 concludes the study with policy implications.

## 2. The Determinants of FDI

This study is eminently empirical; therefore, an extensive summary of FDI theories is beyond its scope. Nevertheless, over the years, numerous econometric studies have been conducted, and different paradigms and theories explaining the determinants of FDI and location choice of multinational companies have emerged. Then again, these theoretical assumptions and models are not reciprocal, and a general theory clarifying the FDI determinants is still inexplicit. Blonigen [3], Faeth [28], and Moosa [29] are some of the major surveys in the econometric literature on the determinants of FDI. Building on these studies, this paper extends our knowledge about the investment climate in CIS member-states and sheds light on the impact of natural resources, economic freedom, and sea-access on inward FDI.

### 2.1. Natural Resources and Inward FDI

Theoretical literature has long established the importance of natural resources in international business. Dunning [30] OLI paradigm is perchance the most renowned theory delineating that firms engage in foreign investment, when the three advantages, [O] ownership, [L] location, and I [internationalization], are achieved [28]. Dunning [30] explains that international firms can reduce intrinsic costs and weaknesses by engaging in foreign investment and gaining control over critical resources that can be used as leverage in the host country. Likewise, resource dependence theory asserts that resources are crucial for organizational success and that uninterrupted access and control over resources collectively promote competitive advantage [31]. Market constraints and uncertainties in the stable flow of critical resources, such as raw materials, are predominant factors affecting FDI location decisions of MNEs [32]. Factor endowments-based trade theory also suggests that FDI flows to countries with a comparative advantage in factor endowments, such as natural resources [33]. Hence, owing to the desire of firms to access low-cost resources under lenient regulations and increase profit, resource endowments are vital for inward FDI.

Natural resource abundance is typically perceived as an advantage used to attract FDI. However, results in empirical literature are inconclusive, with positive impacts [4,10,34] often being countered by adverse outcomes [11,12,35], and there is no joint agreement on the explicit effect of natural resources on FDI. Some empirical studies relate the adverse impact of natural resources on FDI to the crowding-out effects of resource FDI [11,12]. However, Gonchar and Marek [36] emphasize that resource FDI does not crowd out non-resource FDI and suggest that the difference between positive and negative outcomes depends on the measurement criteria chosen for available resources. Alternatively, Kang [10] and Kolstad and Wiig [37] suggest that inconsistent empirical results are contingent on the institutions of both home and host countries. Notably, prodigious contributions of resource FDI to economic growth, poverty reduction, and sustainable development are evident in some resource-rich developing countries, such as Chile, Brazil, Indonesia, Malaysia, and Botswana [16]. Hence, considering that most of the FDI in CIS can be explained by the abundance of oil, natural gas, and minerals, the gravity of natural resources in attracting FDI inflows to the region is inevitable.

### 2.2. Economic Freedom and Inward FDI

Empirical research on the host country's institutions and inward FDI has, in general, revealed the positive influence of the institutional and legal framework [38]. Thus, the pace and specifics of institutional improvement in the host country are critical factors in attracting FDI inflows, especially in transition economies, where entire political, economic, and legal environments have undergone fundamental transformations, following the failure of the ossified socialist economic system. However, regardless of the broad spectrum of evidence that institutions matter, a conjoint consensus on their relative importance is yet to be established [38,39]. The economic freedom index, devised by the Heritage Foundation, assesses countries' economic and business environment from 12 different perspectives and therefore is viewed as a sufficient measure reflecting the domestic business and

investment climate in host countries [18,40]. The relationship between economic freedom and FDI is often delineated through its positive effects [22,41]. Liberalized economies tend to minimize the risks associated with investing in a particular region, reduce transaction costs, ensure property rights protection, promote ease of doing business, permit fair competition, maximize economic return, and encourage inward FDI [42,43]. Besides, open economies with efficient institutions reflect attractive business and investment climate, contribute to economic growth, and promote conditional convergence in developing economies to catch up with developed economies [21]. On the contrary, closed economies increase the costs of doing business, create market uncertainty, and correspondingly deter inward FDI [10]. Therefore, the extent to which foreign investors increase investment flows in a particular location is contingent on the quality and effectiveness of the legal and regulatory institutions.

Over the years, various empirical studies have examined the economic freedom–FDI nexus and found robust evidence that economic freedom is positively linked with FDI inflows [17–19,21,42]. These studies have described economic freedom as an instrument facilitating growth in host economies' absorptive capacity with regard to assimilating know-how and technological spillovers from overseas affiliates of MNEs and thus encouraging FDI inflows. Considering different groups of countries, some studies, such as Bengoa and Sanchez-Robles [17] found a positive association between economic freedom and inward FDI in Latin America. Quazi [18] reported a similar relationship in East Asia. More recent research, Economou [19] and Ghazalian and Amponsem [21], have also arrived at similar conclusions in South Europe and developing countries that the higher degree of economic freedom signposts the presence of attractive domestic business and investment climate. Besides, as higher economic freedom scores are associated with greater investment throughput, as expressed by the influence of foreign investment on economic development, FDI inflows tend to be relatively higher in countries with liberated markets [10]. Therefore, to attract foreign investment flows, host countries should initiate reforms directed to enhancing microeconomic performances with respect to foreign trade, property rights and government intervention, fiscal burden, banking, and finance [20].

## 2.3. Geographic Location and Inward FDI

Despite increasing global economic integration, geographic remoteness and poor transportation infrastructure isolate economies, preventing them from participating in global production networks [24]. Essentially, market access is a determinant of market potential, affecting the export demand for each country based on its geographical location and that of its partners in trade. Country-specific geographic factors have a critical influence on the cost of doing business, including transport costs, the flow of goods, and the production process. Therefore, the geographic location is often the root cause behind the reluctance of multinational companies to move production facilities to geographically disadvantaged countries, despite low-cost labor, natural resource endowments, and institutional quality [23].

The literature presents the use of various geographic measures, including the distance between countries, distance to the core world markets, whether or not countries share a common border, access to the coastlines, and whether states are landlocked or island [24,44]. Distance has long been regarded as a negative factor because it is a source of friction between markets. Hence, high transportation costs or other trade barriers designate that more distant states experience market access forfeit and face additional costs on imports and exports [26,45]. In contrast, whether countries share a common border is often regarded as a cost-reducing factor because neighboring countries usually have integrated transportation networks and are expected to have transit and customs treaties that decrease transit times, shipping costs, and insurance expenses [23,24]. Consistently, some previous studies have corroborated these assertions [46,47]. Numerous studies on FDI and trade have highlighted that territorial coastlines provide a significant geographic location advantage [41,48]. Due to direct access to sea-routes and international waters, countries with sea-access mostly have lower transport costs and stress-free access to international markets [23]. Hence, countries with territorial coastlines are necessarily prosperous in foreign trade, more integrated into the global economy, and absorb higher volumes of inward FDI [48,49]. On the other hand, countries without territorial sea- access are substantially dependent

on the capacity of surrounding transit countries to develop robust infrastructure, build close political ties, and to maintain regional peace and stability [50]. A sudden rise of a conflict between adjacent countries may halt transit movements, increase transportation costs, limit access to foreign markets, impose restrictions on return on investment, and divert FDI inflows due to the associated risk and uncertainty. Despite the importance of physical geography on international trade and investment, research on the impact of geographic location on inward FDI has been rather scarce, with the exceptions of Coughlin and Segev [48], Amiti and Smarzynska [49], and Chanegriha, Stewart and Tsoukis [44].

## 3. Data and Methodology

### 3.1. Description of Data

Contingent upon data available on all variables and countries considered, this study uses a balanced panel dataset over the 20 years, from 1998 to 2017, for 12 CIS member-states. Our sample includes countries from Eastern Europe (Belarus, Russia, Ukraine), Western Asia (Armenia, Azerbaijan, Georgia, Moldova), and Central Asia (Kazakhstan, Kyrgyzstan, Tajikistan, Turkmenistan, Uzbekistan). Though Georgia and Ukraine have officially left the CIS, these countries are discussed in the context of this group of countries due to geographic proximity and similarities in economic structure. Data are obtained from the World Development Indicators of the World Bank, the United Nations Conference on Trade and Development (UNCTAD), and the Heritage Foundation.

Following the preponderance of literature on FDI, [5,19,51,52], our dependent variables are the log of total FDI inflows and FDI per capita. Using the FDI per capita ratio as a dependent variable helps to adjust FDI inflows for country-specific population size and facilitates better comparisons of FDI distribution patterns [53].

### 3.2. Natural Resources

Resource abundance is a significant advantage of a country, influencing inward FDI distribution patters, especially in developing economies [4,54]. Existing studies have used various measures, such as the share of natural resource rents in GDP, fuel as a percentage of merchandise exports, and minerals as a share of merchandise exports, to proxy for natural resources. This study employs four different measures to account for natural resources: (1) revealed comparative advantage (RCA) in petroleum, (2) oil as a percentage of merchandise exports, (3) the share of minerals in merchandise exports, and (4) the share of oil and minerals in merchandise exports. These measures are employed for four reasons. First, revealed comparative advantage in petroleum is a new proxy for natural resources that provides a general indication and approximation of competitive export strength of a country in petroleum, and a higher value implies higher export strength. Based on the Ricardian trade theory, a country is said to possess a revealed comparative advantage in petroleum when its ratio of exports of petroleum to its total exports of all goods (products) exceeds the same ratio for the world as a whole. Qiu [55] and Harding [56] have considered this metric in prior FDI literature. Second, these variables can help to interpret the type of FDI inflows to CIS economies. For instance, oil-exporting member-states are likely to host FDI flows towards the oil extraction sector. Third, these measures interpret the gravity of resource endowments to the host country by capturing the host economy's comparative advantage and dependence on petroleum, oil, and mineral exports. Fourth, these proxies have been used in prior research, and the data are readily obtainable.

### 3.3. Economic Freedom

Economically free countries protect the rights of individuals and companies to have control over their labor and property and freedom to pursue business interests without coercion or restriction of liberty. The economic freedom index provides a broad economic and entrepreneurial environment overview of a country and is extensively used in literature. This composite measure is constructed based on four composite groups of indicators representing the economic and regulatory regime of

a country, the rule of law, government size, regulatory efficiency, and market openness. These four measures are estimated using 12 quantitative and qualitative economic freedom factors, signifying the degree to which the economy is following open-market principles [7]. This index uses a 0–100 scale to estimate the economic openness of a country, where a higher score indicates economic freedom to international business, the availability of stable market institutions, effective monetary policies, and the ease of doing business. Numerous empirical studies, such as Quazi [18] and Godinez and Liu [51], have well documented the positive relationship between economic freedom and inward FDI.

### 3.4. Sea-Access

Existing econometric literature has underlined country-specific geographic location to be a significant factor influencing international trade, investment flows, and economic development [24,44,55]. Hence, given the geographic location heterogeneity of CIS member-states, where some countries are landlocked with much less maneuverability and narrower policy options, while others enjoy the collateral benefits of territorial coastlines, this study accounts for the existence of sea-access. Because geographic proximity to the sea seems to be a crucial factor affecting both foreign policy decisions of host economies and investment location decisions of foreign investors, we group CIS member-states into countries with and without territorial access to coastlines. Based on their geographic location, Kazakhstan, Azerbaijan, and Turkmenistan share the shores of the Caspian Sea with Iran and Russia, whereas Russia, Georgia, and Ukraine share the coastlines of the Black Sea with Bulgaria, Moldova, Romania, and Turkey. Since the countries in our sample are classified into two, with and without territorial coasts, we construct one dummy variable and denote it as 0 and 1, where "sea-access" is equal to 1, when a given country possesses territorial coastlines, and 0 if otherwise.

### 3.5. Control Variables

Consistent with the majority of existing literature, we use market size, trade openness, external debt, exchange rate, and infrastructure as control variables. Empirically, numerous studies have shown a substantially strong relationship between these variables and FDI in terms of both statistical significance and casualty relationship [56,57]. The host country's market size is critical for MNEs seeking to expand their operations abroad to reap substantial profits. Besides, larger markets offer more prospects for firms to achieve economies of scale and reduce production-related costs. Hence, sizeable markets are bound to attract more foreign companies and enhance market competitiveness [58]. The majority of the existing studies have used GDP as a standard measure for market size, and a positive relationship between the market size and inward FDI has been well documented. Trade (exports + imports) to GDP ratio is another proxy often used to reflect the host economy's openness to foreign trade. Prior research has established a positive and significant link between the variables, where trade openness affects the nature and volume of FDI [59]. Since the restrictions on trade imposed by host economies directly reflect on their export and import intensity [60], lower trade volume usually indicates unfavorable business climate for MNEs pursuing liberalized economies with minimum trade barriers [5]. Foreign debt is typically perceived to deter inward FDI, and that accumulation of external debt is associated with immense macroeconomic risks and volatility [61]. Besides, external debt is viewed as an alternative source of capital financing [62] and that an increase in foreign debt ought to decrease inward FDI. Hence, the ratio of external debt stocks to GDP is used to proxy for external debt, accounting for market risk and instability [63]. The exchange rate instability in a host country creates uncertainties regarding the future activities of MNEs, which, consequently, may decrease FDI inflows [64]. The official exchange rate is often used to measure exchange rate risk and is found to be a detrimental factor in various studies [5,52]. As for the risk aversion theory, exchange rate instability deters FDI [65], whereas a stable exchange rate may encourage it. Infrastructure, on the other hand, is a broad concept covering various dimensions, including roads, railways, seaports, and information telecommunication systems affecting transaction costs faced by MNEs [66]. Therefore, regardless of the foreign investment type, the availability of proper infrastructure, enabling the production and distribution of products and services, is critical

to foreign investors [11]. Following Kolstad and Wiig [34] and Mina [67], we employ mobile cellular subscriptions per 100 people to proxy for telecommunications infrastructure development.

The variables are chosen based on the prior FDI literature but constrained by data availability in some cases. For instance, data on bilateral FDI, tax legislation, trade policies, and real wages are not readily available for all CIS states, particularly for Tajikistan, Turkmenistan, and Uzbekistan. Table 1 provides a complete list of variables, including the scales of measurement, expected effect, and data sources. Table 2 reports the descriptive statistics, and the correlation matrix is presented in Table 3.

**Table 1.** Description of Variables, Expected Effect, and Data Sources.

| Variables | Description | Expected Effect | Data Sources |
|---|---|---|---|
| FDI inflows | Log of FDI inflows in current US dollars | | World Bank |
| FDI per capita | Log of FDI per capita in current US dollars | | UNCTAD |
| RCA Petroleum | Revealed comparative advantage in petroleum | + | UNCTAD |
| Oil Exports | Oil % of total merchandise exports, % | + | World Bank |
| Minerals Exports | Minerals % of total merchandise exports, % | + | World Bank |
| Oil and Minerals | Oil and Minerals % of total merchandise exports, % | + | World Bank |
| Economic Freedom | Economic freedom index, (0–100) | + | Heritage Foundation |
| Sea-Access | A dummy variable, 1 if Sea-Access, 0 if otherwise | + | Britannica Encyclopedia |
| Market Size | Log of GDP in current US dollars | + | World Bank |
| Trade Openness | Exports and imports (Trade) to GDP ratio, % | + | World Bank |
| External Debt | Log of total external debt stocks/GDP, % | - | World Bank |
| Exchange Rate | Log of nominal exchange rate (local currency to USD) | - | World Bank |
| Infrastructure | Log of mobile phone subscribers per 100 people | + | World Bank |

**Table 2.** Summary Statistics.

| Variables | Full Sample | | Sea-Access | | No Sea-Access | | Mean Difference |
|---|---|---|---|---|---|---|---|
| | Mean | Std. Dev. | Mean | Std. Dev. | Mean | Std. Dev. | (Sea−No Sea) |
| FDI Inflows | 20.472 | 1.891 | 21.689 | 1.546 | 19.254 | 1.343 | −2.435 *** |
| FDI Per Capita | 4.292 | 1.530 | 5.08 | 1.222 | 3.504 | 1.397 | −1.576 *** |
| RCA Petroleum | 5.175 | 6.321 | 8.233 | 6.966 | 2.116 | 3.588 | −6.117 *** |
| Oil/Exports | 23.159 | 19.82 | 26.809 | 21.632 | 2.391 | 4.514 | −24.418 *** |
| Minerals/Exports | 14.60 | 9.686 | 3.826 | 4.227 | 9.729 | 12.37 | 5.903 *** |
| Oil and Minerals/Exports | 6.778 | 19.956 | 30.636 | 20.768 | 12.12 | 13.988 | −18.516 *** |
| Economic Freedom | 53.592 | 8.938 | 53.492 | 8.697 | 53.691 | 9.207 | 0.198 |
| Market Size | 23.688 | 1.771 | 24.599 | 1.786 | 22.777 | 1.199 | −1.823 *** |
| Trade Openness | 92.291 | 30.35 | 85.633 | 25.538 | 98.948 | 33.293 | 16.623 *** |
| External Debt | 19.228 | 1.816 | 19.846 | 2.18 | 18.61 | 1.052 | −1.236 *** |
| Exchange Rate | 3.346 | 2.627 | 1.935 | 1.847 | 4.758 | 2.536 | 2.822 *** |
| Infrastructure | 2.913 | 2.391 | 3.252 | 2.118 | 2.574 | 2.601 | −0.678 ** |
| Number of Observations | 240 | | 120 | | 120 | | |

Note: ***, **, * show significance at the 0.01, 0.05, and 0.10 levels.

**Table 3.** Correlation Matrix.

| Variables | 1 | 2 | 3 | 4 | 5 | 6 | 7 | 8 | 9 | 10 | 11 | 12 | 13 |
|---|---|---|---|---|---|---|---|---|---|---|---|---|---|
| FDI Inflows | 1.000 | | | | | | | | | | | | |
| FDI Per Capita | 0.804 * | 1.000 | | | | | | | | | | | |
| RCA Petroleum | 0.392 * | 0.301 * | 1.000 | | | | | | | | | | |
| Oil/Exports | 0.524 * | 0.432 * | 0.695 * | 1.000 | | | | | | | | | |
| Mineral/Exports | 0.016 | 0.012 | −0.314 * | −0.230 * | 1.000 | | | | | | | | |
| Oil & Mineral/Exports | 0.528 * | 0.423 * | 0.537 * | 0.881 * | 0.256 * | 1.000 | | | | | | | |
| Economic Freedom | 0.094 | 0.387 * | −0.227 * | −0.092 | 0.044 | −0.221 * | 1.000 | | | | | | |
| Sea−Access | 0.645 * | 0.516 * | 0.485 * | 0.571 * | 0.514 * | 0.521 * | −0.011 | 1.000 | | | | | |
| Market Size | 0.869 * | 0.470 * | 0.324 * | 0.281 * | 0.487 * | 0.526 * | −0.094 | 0.516 * | 1.000 | | | | |
| Trade Openness | −0.310 * | −0.093 | −0.020 | −0.065 | −0.166 | −0.285 * | −0.095 | −0.220 * | −0.417 * | 1.000 | | | |
| External Debt | 0.678 * | 0.281 * | 0.155 * | 0.033 | 0.377 * | 0.273 * | 0.137 * | 0.341 * | 0.842 * | −0.436 * | 1.000 | | |
| Exchange Rate | −0.010 | −0.122 | −0.115 | −0.314 * | −0.099 | −0.037 | −0.027 | −0.538 * | 0.195 * | −0.104 | 0.261 * | 1.000 | |
| Infrastructure | 0.608 * | 0.657 * | −0.036 | 0.032 | 0.150 * | 0.211 * | 0.443 * | 0.142 * | 0.517 * | −0.195 * | 0.452 * | 0.128 * | 1.000 |

Note: * shows significance at the 0.05 and 0.01 levels.

### 3.6. Empirical Model Specification

This study employs panel corrected standard errors (PCSEs) regression developed by Beck and Katz [68] for all estimations. PCSEs estimator suits best to small panels and accounts for finite sample bias while producing panel-corrected standard errors that allow heteroskedasticity and correlation within panels [69]. Many empirical studies, such as [13,70–72], have employed PCSEs estimator and highlighted that for data sets with features as in this research, PCSEs offers a better fit and rather robust estimates than other alternative models. Nevertheless, to test the robustness of regression results to changes in the underlying econometric model, we augment PCSE estimations with Parks [73] Feasible Generalized Least Squares (FGLS), Hoechle [74] GLS Random Effects with Driscoll-Kraay standard errors, and Baltagi and Wu [75] GLS estimator of the Random Effects (RE). The following equation is estimated using Beck and Katz [68] PCSE estimation method:

$$FDI_{it} = \beta_0 + \beta_1 Resources_{i,t} + \beta_2 Economic\ Freedom_{i,t} + \beta_3 Sea\ Access_{i,t} + \beta_4 Control_{i,t} + \varepsilon_{i,t}$$

For $i = 1, \ldots, N$ is the number of panels; $t = 1, \ldots, T_i$; $T_i$ is the number of periods in panel $i$, and $FDI_{i,t}$ is the dependent variable, representing the log of net FDI inflows and the log of FDI per capita to country $i$ in period $t$. $Resources_{i,t}$, $Institutions_{i,t}$, and $SeaAccess_{i,t}$ are the main explanatory variables signifying natural resources, economic freedom, and availability of sea-access, whereas $Control_{i,t}$ denotes a group of control variables that vary over $t$ and $i$, and $\varepsilon_{it}$ is the idiosyncratic error term. To control for potential autocorrelation within panels, we specify our model for first-order autocorrelation AR (1) with a common coefficient [69] and obtain Prais–Winsten parameter estimates.

## 4. Empirical Results

Table 4 presents Prais–Winsten regression estimates with panels corrected standard errors robust to cross-sectional dependence, autocorrelation, and heteroskedasticity for two dependent variables, total FDI inflows and FDI per capita. The results indicate that our main explanatory variables, natural resources, economic freedom, and sea-access, are positive and statistically significant, encouraging FDI inflows to CIS countries.

Model 1 presents regression estimates for control variables. The regression results show that market size and communications infrastructure are positive and statistically significant ($p < 0.01$ for market size; $p < 0.01$ for infrastructure), while the exchange rate is negative and significant ($p < 0.01$). These outcomes suggest that member-states with larger market size and better telecommunications infrastructure are likely to receive more FDI, while economies with higher exchange rates tend to receive less FDI. Prior studies have also found that sizeable markets with developed infrastructure offer better opportunities for MNEs to achieve higher economies of scale.

Models 2–4 provide separate regression estimates for each main explanatory variable. The results in Model 2 show a positive and significant association between natural resources and inward FDI ($p < 0.01$), indicating that resources are substantial factors encouraging FDI flows to CIS. This outcome signposts the existence of resource-seeking FDI in CIS countries and supports the factor endowment theory that when a country has a comparative advantage in factor endowment, such as natural resources, then foreign investment flows to that country are mainly driven by this factor endowment.

Model 3 reveals that economic freedom is positively and significantly associated with inward FDI to CIS member-states ($p < 0.01$). This relationship signifies that a higher degree of economic liberalization encourages inward FDI, implying that reforms towards a market economy, built around less-restrictive business regulations and free international trade principles, can be auspicious for inward FDI location in CIS transition economies.

**Table 4.** Prais-Winsten regression with PCSEs.

| Dependent Variable: | Total FDI Inflows | | | | | FDI Per Capita | | | | |
|---|---|---|---|---|---|---|---|---|---|---|
| | **Model (1)** | **Model (2)** | **Model (3)** | **Model (4)** | **Model (5)** | **Model (1)** | **Model (2)** | **Model (3)** | **Model (4)** | **Model (5)** |
| Natural Resource | | | | 0.032 *** −3.3 | 0.021 ** −2.36 | | | | 0.032 ** −2.59 | 0.028 *** −2.65 |
| Economic Freedom | | | 0.026 *** −2.55 | | 0.021 ** −2.45 | | | 0.046 *** −3.44 | | 0.043 *** −3.92 |
| Sea−Access | | 1.256 *** −6.5 | | | 0.976 *** −5.15 | | 1.536 *** −4.86 | | | 1.180 *** −4.4 |
| Market Size | 0.948 *** −10.51 | 0.693 *** −8.96 | 1.084 *** −10.97 | 0.878 *** −9.67 | 0.802 *** −8.98 | 0.429 *** −3.1 | 0.136 −1.14 | 0.650 *** −4.97 | 0.394 *** −2.83 | 0.340 *** −2.72 |
| Trade Openness | 0.004 −1.26 | 0.005* −1.82 | 0.003 −1.25 | 0.003 −1.13 | 0.004 * −1.69 | 0.004 −1.02 | 0.004 −1.12 | 0.002 −0.54 | 0.006 ** −2.04 | 0.003 −0.84 |
| Foreign Debt | −0.096 (−1.27) | −0.074 (−1.16) | −0.195 ** (−2.40) | −0.08 (−1.09) | −0.137 ** (−2.07) | −0.202 * (−1.67) | −0.223 ** (−2.36) | −0.406 *** (−3.48) | −0.213 * (−1.87) | −0.363 *** (−3.80) |
| Exchange Rate | −0.130 *** (−3.81) | 0.022 −0.55 | −0.121 *** (−4.14) | −0.116 *** (−3.57) | 0.003 −0.09 | −0.137 ** (−2.50) | 0.056 −0.92 | −0.111 *** (−2.87) | −0.118 ** (−2.28) | 0.043 −0.83 |
| Infrastructure | 0.157 *** −3.66 | 0.188 *** −4.72 | 0.097 ** −2.1 | 0.182 *** −4.34 | 0.150 *** −3.37 | 0.295 *** −4.59 | 0.346 *** −5.99 | 0.215 *** −3.43 | 0.324 *** −5.12 | 0.271 *** −4.3 |
| Constant | −0.511 (−0.38) | 3.808 *** −3.13 | −3.053** (−2.01) | 0.61 −0.47 | 1.58 −1.08 | −2.742 (−1.27) | 3.093 −1.52 | −6.194 *** (−3.12) | −1.595 (−0.78) | −0.978 (−0.47) |
| Number of Observations | 240 | 240 | 240 | 240 | 240 | 240 | 240 | 240 | 240 | 240 |
| R-Squared | 0.874 | 0.873 | 0.865 | 0.873 | 0.869 | 0.261 | 0.386 | 0.378 | 0.303 | 0.477 |
| Wald Chi-Squared | 524.56 *** | 770.57 *** | 654.90 *** | 644.69 *** | 960.95 *** | 74.09 *** | 144.15 *** | 131.77 *** | 89.98 *** | 226.57 *** |

Note: The *T*−statistics are in parentheses, *** $p < 0.01$, ** $p < 0.05$, * $p < 0.1$.

The estimates in Model 4 show that FDI inflows to CIS countries are sensitive to physical geographic location, as expressed by the statistically significant and positive coefficient for sea-access ($p < 0.01$). This result demonstrates the comparative advantage of countries with sea-access over their counterparts without territorial coastlines and documents that lack of sea access can impose critical constraints on inward FDI.

Model 5 is the benchmark regression with the three main explanatory variables. The estimates indicate that our main variables of interest remain positive and statistically significant ($p < 0.05$ for natural resource; $p < 0.05$ economic freedom; $p < 0.01$ for sea-access). Holding all of the other independent variables fixed, a unit percentage change in revealed comparative advantage in petroleum is expected to result in a 2.1 percent change in FDI inflows. Equally, one unit increase in the economic freedom index is expected to grow FDI inflows by 2.1 percent. In contrast, hypothetical territorial access to the sea tends to increase FDI inflows to countries without sea-access by 165 percent. When the dependent variable is log(y) and $\hat{\beta}_1$ is the coefficient of a time-invariant variable, then the exact percentage difference in the predicted y for $x_1 = 1$ versus when $x_1 = 0$ can be estimated using $100 * \left[\exp\left(\hat{\beta}_1\right) - 1\right]$. Hence, the relative impact of sea-access is: $100*[\exp. (0.976) -1] = 100*(1.654) \approx 165\%$. These results are consistent for both dependent variables, FDI inflows, and FDI inflows per capita, reflecting the importance of resource endowments, economic freedom, and territorial sea-access in attracting FDI to CIS member-states.

In Table 5, we re-estimate the Model 5 (Table 4) using alternative measures on natural resources. In Model 1, we use the share of oil in merchandise exports. The estimates indicate that the coefficient on oil is statistically significant and positive for both dependent variables ($p < 0.05$). Specifically, holding all other variables constant, one percent unit change in the share of oil in exports leads to 0.9 and 0.4 percent change in FDI inflows and FDI per capita, respectively. This result confirms that oil is a significant factor in attracting FDI inflows to resource-abundant CIS economies, such as Azerbaijan, Kazakhstan, Russia, Turkmenistan, and Uzbekistan. In Model 2, we employ the share of minerals in total merchandise exports, but the relationship is not statistically significant. However, using the share of oil and minerals in exports in Model 3, we find that the coefficient for oil and minerals is associated with an average increase of 0.8 in FDI inflows when controlled for other variables. Nevertheless, although positive, this result is only significant at the 10 percent level for FDI per capita.

Our empirical analyses proceed by examining the impact of four aspects of economic freedom, the rule of law, government size, regulatory efficiency, and open markets, on inward FDI to CIS in Table 6. The regression estimates show that two economic freedom sub-components, government size, and open markets, are positively related to FDI inflows to CIS ($p < 0.01$ for government size and $p < 0.05$ for open markets). Specifically, a one percent unit increase in government size results in, on average, a 2.9 percent increase in FDI inflows. In comparison, a similar change in open markets increases FDI inflows by 2.1 percent, on average, holding the other factors constant. These findings signify that lower tax burden and government spending, along with higher freedom on trade, investment, and finance, can encourage FDI to CIS states. Prior literature [19,21] has also accentuated that taxes and government spending are unavoidable burdens that may cause budget deficits and public debt accumulation, which collectively inhibit economic dynamism. Barriers to trade, constraints on investment, and strict government regulations and interference in the financial sector hinder trade, impede domestic and foreign capital flows, and reduce access to credit that jointly deter the host country's investment climate [17]. Hence, lower tax burdens, government spending, tariff as well as non-tariff barriers to trade, regulatory restrictions on capital flows, and government regulations and interventions in the financial sector would create an enticing business environment affecting FDI distribution patterns in the corresponding host countries.

**Table 5.** Prais-Winsten regression with PCSEs.

| Dependent Variable: | Total FDI Inflows | | | FDI Per Capita | | |
|---|---|---|---|---|---|---|
| | Oil % Exports | Min % Exports | Oil & Min Exports | Oil % Exports | Min % Exports | Oil & Min Exports |
| Natural Resource | 0.009 ** | 0.005 | 0.008 ** | 0.004 ** | −0.005 | 0.004 * |
| | −2.28 | −0.56 | −2.13 | −2.13 | (−0.45) | −1.69 |
| Economic Freedom | 0.018 *** | 0.017 ** | 0.018 *** | 0.037 *** | 0.039 *** | 0.037 *** |
| | −2.37 | −2.01 | −2.33 | −3.5 | −3.47 | −3.65 |
| Sea-Access | 1.010 *** | 1.181 *** | 1.056 *** | 1.312 *** | 1.397 *** | 1.348 *** |
| | −5.87 | −6.01 | −5.78 | −5.61 | −5.18 | −5.28 |
| Market Size | 0.749 *** | 0.811 *** | 0.775 *** | 0.311 *** | 0.320 *** | 0.329 ** |
| | −8.11 | −8.13 | −8.85 | −2.36 | −2.38 | −2.54 |
| Trade Openness | 0.006 ** | 0.005 * | 0.006 *** | 0.004 | 0.003 | 0.003 |
| | −2.56 | −1.88 | −2.66 | −1.21 | −0.82 | −1.07 |
| Foreign Debt | −0.094 | −0.146 ** | −0.113 * | −0.349 *** | −0.350 *** | −0.360 *** |
| | (−1.52) | (−2.06) | (−1.81) | (−3.70) | (−3.50) | (−3.62) |
| Exchange Rate | 0.018 | 0.013 | 0.012 | 0.066 | 0.062 | 0.061 |
| | −0.51 | −0.32 | −0.34 | −1.31 | −1.12 | −1.17 |
| Infrastructure | 0.151 *** | 0.137 *** | 0.136 *** | 0.267 *** | 0.273 *** | 0.262 *** |
| | −3.52 | −2.98 | −3.23 | −4.34 | −4.26 | −4.26 |
| Constant | 1.954 | 1.601 | 1.62 | −0.602 | −0.53 | −0.713 |
| | −1.33 | −0.95 | −1.12 | (−0.28) | (−0.23) | (−0.33) |
| Number of Observations | 240 | 240 | 240 | 240 | 240 | 240 |
| R-Squared | 0.862 | 0.871 | 0.863 | 0.495 | 0.455 | 0.47 |
| Wald Chi-Squared | 1185.65 *** | 919.36 *** | 1161.98 *** | 261.30 *** | 212.86 *** | 236.70 *** |

Note: The *T*-statistics are in parentheses, *** $p < 0.01$, ** $p < 0.05$, * $p < 0.1$.

**Table 6.** Prais-Winsten regression with PCSEs.

| Dependent Variable: | Total FDI Inflows | | | | FDI Per Capita | | | |
|---|---|---|---|---|---|---|---|---|
| | Rule of Law | Government Size | Regulatory Efficiency | Open Markets | Rule of Law | Government Size | Regulatory Efficiency | Open Markets |
| Natural Resource | 0.017 * | 0.021 ** | 0.019 ** | 0.020 ** | 0.019 * | 0.023 ** | 0.024 ** | 0.027 ** |
| | −1.84 | −2.44 | −2.09 | −2.15 | −1.75 | −2.16 | −2.29 | −2.44 |
| Economic Freedom | −0.008 | 0.027 *** | 0.005 | 0.018** | −0.004 | 0.031 *** | 0.011 | 0.024 *** |
| | (−0.92) | −3.84 | −0.74 | −2.13 | (−0.42) | −3.26 | −1.36 | −3.72 |
| Sea-Access | 1.152 *** | 0.820 *** | 1.100 *** | 1.072 *** | 1.407 *** | 1.044 *** | 1.440 *** | 1.307 *** |
| | −6.34 | −4.3 | −6.36 | −5.68 | −4.71 | −3.23 | −5.64 | −4.79 |
| Market Size | 0.650 *** | 0.780 *** | 0.707 *** | 0.736 *** | 0.105 | 0.228 * | 0.15 | 0.271 ** |
| | −7.66 | −10.43 | −8.48 | −8.58 | −0.84 | −1.89 | −1.33 | −2.22 |
| Trade Openness | 0.005 * | 0.009 *** | 0.004 * | 0.004 | 0.004 | 0.008 ** | 0.003 | 0.003 |
| | −1.77 | −3.09 | −1.75 | −1.57 | −1.07 | −2.07 | −1.03 | −0.77 |
| Foreign Debt | −0.039 | −0.012 | −0.085 | −0.102 | −0.204 ** | −0.154 * | −0.263 *** | −0.326 *** |
| | (−0.54) | (−0.21) | (−1.30) | (−1.53) | (−2.00) | (−1.70) | (−3.04) | (−3.27) |
| Exchange Rate | 0.018 | −0.004 | 0.013 | 0.013 | 0.051 | 0.029 | 0.065 | 0.059 |
| | −0.47 | (−0.12) | −0.36 | −0.34 | −0.88 | −0.5 | −1.31 | −1.07 |
| Infrastructure | 0.202 *** | 0.139 *** | 0.180 *** | 0.181 *** | 0.363 *** | 0.290 *** | 0.332 *** | 0.317 *** |
| | −4.98 | −3.56 | −3.89 | −4.26 | −6.13 | −4.79 | −5.4 | −5.32 |
| Constant | 4.298 *** | −1.484 | 3.447 ** | 3.012 ** | 3.496 * | −2.762 | 2.824 | 0.807 |
| | −3.6 | (−0.83) | −2.59 | −2.27 | −1.74 | (−1.02) | −1.48 | −0.4 |
| Number of Observations | 240 | 240 | 240 | 240 | 240 | 240 | 240 | 240 |
| R-Squared | 0.873 | 0.874 | 0.867 | 0.873 | 0.403 | 0.439 | 0.471 | 0.438 |
| Wald Chi-Squared | 830.49 *** | 956.66 *** | 919.29 *** | 845.12 *** | 148.90 *** | 163.42 *** | 201.62 *** | 216.12 *** |

Note: *Rule of Law*—property rights and government integrity, *Government Size*—government spending and tax burden, *Regulatory Efficiency*—business freedom and monetary freedom, and *Open Markets*—trade freedom, investment freedom, and financial freedom. The *T*−statistics are in parentheses, *** $p < 0.01$, ** $p < 0.05$, * $p < 0.1$.

*Robustness Tests*

To test the robustness of results in the benchmark regression to changes in the underlying empirical model, we employ three alternative estimators, namely FGLS, RE (with Driscoll-Kraay

standard errors), and RE (GLS), in Table 7. From the estimates reported, it is evident that the main independent variables are significant and positively associated with both dependent variables across all models. Holding other independent variables fixed, we may expect, on average, a 2 percent increase in total FDI inflows, and 2.5 percent in FDI per capita for a one percent unit increase in natural resources. Likewise, one unit increase in economic freedom index leads to, on average, a 1.8 percent increase in total FDI inflows and a 3.6 percent increase in FDI per capita. The coefficients for sea-access imply that countries with coastlines receive, on average, 158 percent and 168 percent more inward FDI than that of their counterparts without territorial coastlines.

**Table 7.** Robustness Test: FGLS, RE (D-K), RE (GLS).

| Dependent Variable: | Total FDI Inflows | | | FDI Per Capita | | |
|---|---|---|---|---|---|---|
| | **FGLS** | **RE (D-K)** | **RE (GLS)** | **FGLS** | **RE (D-K)** | **RE (GLS)** |
| Natural | 0.020 *** | 0.019 ** | 0.020 ** | 0.030 *** | 0.020 ** | 0.020 ** |
| Resource | −2.98 | −2.12 | −2.02 | −4.45 | −2.35 | −2.4 |
| Economic | 0.018 *** | 0.017 ** | 0.021 ** | 0.048 *** | 0.028 ** | 0.032 *** |
| Freedom | −3.79 | −2.32 | −2.25 | −8.24 | −3.68 | −2.71 |
| Sea-Access | 0.949 *** | 0.950 *** | 0.945 *** | 1.217 *** | 0.883** | 0.859** |
| | −8.01 | −5.41 | −3.73 | −8.8 | −2.71 | −2.3 |
| Market Size | 0.795 *** | 0.842 *** | 0.820 *** | 0.319 *** | 0.490 *** | 0.420 *** |
| | −14.95 | −12.04 | −8.07 | −4.88 | −3.85 | −3.34 |
| Trade | 0.004 *** | 0.005 ** | 0.004 * | 0.002 | 0.007 *** | 0.005 ** |
| Openness | −2.66 | −2.7 | −1.82 | −1.25 | −3.46 | −2.22 |
| Foreign Debt | −0.129 *** | −0.181 *** | −0.158 ** | −0.361 *** | −0.209 *** | −0.296 *** |
| | (−2.81) | (−5.39) | (−2.06) | (−6.12) | (−4.44) | (−3.29) |
| Exchange | 0.013 | −0.005 | 0.003 | 0.074 * | −0.042* | −0.022 |
| Rate | −0.44 | (−0.14) | −0.09 | −1.82 | (−1.91) | (−0.39) |
| Infrastructure | 0.167 *** | 0.179 *** | 0.152 *** | 0.290 *** | 0.264 *** | 0.276 *** |
| | −6.5 | −3.91 | −4 | −8.85 | −5.52 | −6.24 |
| Constant | 1.716 ** | 1.522 | 1.39 | −0.857 | −4.960 *** | −3.45 |
| | −2.04 | −1.09 | −0.77 | (−0.71) | (−3.65) | (−1.52) |
| Number of Observations | 240 | 240 | 240 | 240 | 240 | 240 |
| R-Squared | | 0.873 | 0.878 | | 0.604 | 0.697 |
| Wald Chi-Squared | 2286.09 *** | 813.79 *** | 787.05 *** | 874.64 *** | 577.85 *** | 243.17 *** |

Note: The *t*-statistics are in parentheses, *** $p < 0.01$, ** $p < 0.05$, * $p < 0.1$.

To account for potential endogeneity concerns, this study employs IV (2SLS), LIML, and IV (EC2SLS) estimators in Table 8. The IV (2SLS) and LIML estimators use Lewbel's (2012) method to generate heteroskedasticity-based instruments allowing the identification of structural parameters in regressions with endogenous and mismeasured regressors when traditional external instruments are absent [76]. The LIML method is consistent under a wider variety of asymptotics than IV or 2SLS and has properties attributable to address issues related to small-sample bias and weak instruments [77]. The EC2SLS method uses Baltagi's random-effects estimator and provides a broader range of instruments that can yield gains in small-sample efficiency [78]. These estimators are widely used in econometric studies to fit panel-data models with endogenous covariates [79].

**Table 8.** Endogeneity Test: IV (2SLS), LIML, and IV (EC2SLS).

| Dependent Variable: | Total FDI Inflows | | | FDI Per Capita | | |
|---|---|---|---|---|---|---|
| | IV (2SLS) | LIML | IV (EC2SLS) | IV (2SLS) | LIML | IV (G2SLS) |
| Natural Resource | 0.028 *** | 0.028 *** | 0.026** | 0.038 *** | 0.038 *** | 0.036 *** |
| | −3.07 | −3.08 | −2.77 | −4.55 | −4.55 | −3.74 |
| Economic Freedom | 0.036 *** | 0.036 *** | 0.035 *** | 0.071 *** | 0.072 *** | 0.064 *** |
| | −4.8 | −4.8 | −3.96 | −8 | −8 | −7.33 |
| Sea-Access | 0.839 *** | 0.838 *** | 0.851 *** | 1.164 *** | 1.161 *** | 1.650 *** |
| | −5.01 | −5 | −2.96 | −5.82 | −5.8 | −7.88 |
| Market Size | 0.880 *** | 0.881 *** | 0.889 *** | 0.446 *** | 0.448 *** | 0.380 *** |
| | −10.97 | −10.96 | −9.25 | −4.7 | −4.71 | −3.02 |
| Trade Openness | 0.004 ** | 0.004 ** | 0.005 ** | 0.006 *** | 0.006 *** | 0.006 *** |
| | −2.21 | −2.22 | −2.55 | −2.72 | −2.73 | −3.5 |
| Foreign Debt | −0.186 *** | −0.187 *** | −0.198 *** | −0.436 *** | −0.437 *** | −0.385 *** |
| | (−3.98) | (−3.98) | (−3.06) | (−7.66) | (−7.66) | (−6.21) |
| Exchange Rate | −0.003 | −0.003 | −0.012 | 0.065 ** | 0.065 ** | 0.063 *** |
| | (−0.12) | (−0.12) | (−0.28) | −2.1 | −2.1 | −4.2 |
| Infrastructure | 0.144 *** | 0.144 *** | 0.159 *** | 0.264 *** | 0.263 *** | 0.202 *** |
| | −4.44 | −4.42 | −4.66 | −6.24 | −6.2 | −3.69 |
| Constant | −0.002 | −0.028 | 0.188 | −4.001** | −4.055 ** | −0.97 |
| | −0.01 | (−0.02) | −0.1 | (−2.21) | (−2.23) | (−0.58) |
| Number of Observations | 228 | 228 | 228 | 228 | 228 | 228 |
| R-Squared | 0.878 | 0.878 | 0.877 | 0.732 | 0.732 | 0.709 |
| F-Statistics | 173.46 | 173.4 | 103.08 | 72.52 | 72.4 | 57.564 |
| Hansen J Statistic | 9.793 (0.200) | 9.781 (0.201) | | 11.419 (0.121) | 11.408 (0.121) | |
| Endogeneity Test | 5.132 (0.023) | 5.132 (0.023) | | 5.557 (0.018) | 5.557 (0.018) | |

Note: Hansen J Statistic: H0: over-identification restrictions are valid (*p*-values are in parentheses). Endogeneity Test: H0: specified endogenous variables can be treated as exogenous (*p*-values are in parentheses). The *t* – statistics are in parentheses, *** $p < 0.01$, ** $p < 0.05$, * $p < 0.1$.

We estimate the benchmark regression using one year lagged value (*t-1*) of the supposed endogenous variable, economic freedom, and as an instrument. The Hansen's J statistic test for over-identification confirms that over-identification restrictions are valid under the null hypothesis (2SLS and LIML estimators use Lewbel's (2012) method to generate other heteroskedasticity-based instruments). The test of endogeneity shows that the null hypothesis of specified endogenous variable can be treated as exogenous is rejected at $p < 0.05$. Many prior studies have employed the lagged value of supposed endogenous covariate as an appropriate instrument [13,80]. Table 8 shows that our main results in benchmark estimation (Model 5 in Table 4) remain consistent across all three estimators. The natural resources, economic freedom, and sea-access have positive and significant coefficients ($p < 0.01$). The coefficients range from 0.026 to 0.027 for natural resource, 0.032 to 0.033 for economic freedom, and 0.850 to 0.880 for sea-access for total FDI inflows. For FDI per capita, the coefficients range from 0.036 to 0.037 for natural resource, 0.065 to 0.069 for economic freedom, and 1.162 to 1.204 for sea-access. These estimates indicate that natural resource abundance, higher economic freedom, and geographic location advantages can indeed encourage FDI to CIS member-states, even after accounting for endogeneity problems.

## 5. Conclusions

This study analyzed the impact of natural resources, economic freedom, and sea-access on FDI inflows using a panel of 12 transition economies in CIS from 1998 to 2017. Our estimates revealed that

natural resources characterized by petroleum and oil reserves, economic freedom, and sea-access are significant factors affecting FDI location decisions of MNEs in CIS.

The empirical evidence suggests that natural resources, particularly oil endowments, encourage FDI flows to CIS. The positive and significant estimates for the revealed comparative advantage in petroleum and the share of oil in merchandise exports corroborate this result. The regression results for economic freedom and its sub-components confirm the hypothesis that liberalized economies attract higher FDI inflows. Specifically, government size (measured by government spending and tax burden) and open markets (measured as trade freedom, investment freedom, and financial freedom) have a robust positive impact on inward FDI. Likewise, the estimates for territorial sea-access also exhibits a positive and significant effect on FDI, implying that countries with territorial coastlines tend to have geographic location advantage and receive substantially more FDI inflows than their counterparts without sea-access. Moreover, larger market size, active participation in global trade, and developing reliable telecommunications infrastructure have a positive and significant influence on FDI inflows to CIS, while the impact of increased external debt is negative and significant.

These results have important policy implications. First, CIS member-states should pursue effective policy initiatives to reap more benefits from resource-seeking FDI. Further, these countries should encourage FDI into the non-resource sector and decrease resource dependence through diversification. These measures help to mitigate the risks associated with the depletion of resource rents and fluctuation of oil prices, which may make large investments in the extractive sector less desirable and lead to an overall decline in inward FDI.

Second, member-states should improve economic freedom, accentuating lower tax burden and government spending as well as higher trade, investment, and financial freedom. These policies should be tailored based on transparency and equity, encouraging innovation and healthy market competition and providing more incentives for businesses to retain and manage a larger share of their income and wealth. Beyond those, these measures should spur private investment and enable firms to pursue their financial goals, expand their economic activities, increase productivity, and create more employment opportunities, thereby contribute to sustainable economic growth.

Third, countries without territorial sea-access, Armenia, Georgia, Kyrgyzstan, Moldova, Tajikistan, and Uzbekistan, should adopt concrete policies aimed at reducing the tyranny of geographic location disadvantage. These states should develop effective microeconomic strategies to offer more benefits enticing to foreign investors and establish close ties with the neighboring transit states because the cost and efficiency of international trade for these economies might be subject to the macroeconomic policies of neighboring countries.

Last but not least, to encourage foreign investment and attain sustained economic development, CIS states should devote resources to boosting economic growth. Besides, member-states should establish a dynamic presence in global trade, enhance telecommunications infrastructure, and reduce reliance on foreign debt and utilize the loans obtained vigilantly.

**Author Contributions:** I.K. (Ibrokhim Karimov) and Y.A. assisted in data curation; W.L. conceived the original idea and supervised the research; I.K. (Ikboljon Kasimov) wrote the original draft and conducted data analysis. All authors have read and agreed to the published version of the manuscript.

**Funding:** This research has received no external funding.

**Acknowledgments:** The authors thank the editor, assistant editor, and the three anonymous reviewers for their constructive comments and suggestions to enhance this manuscript.

**Conflicts of Interest:** The authors declare no conflict of interest.

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
