# Peer review of "Foreign Direct Investment, Natural Resources, Economic Freedom, and Sea-Access: Evidence from the Commonwealth of Independent States"

_sustainability, doi:10.3390/su12083135_

Round 1

Reviewer 1 Report

Comments to the Authors

The paper is focused on Foreign Direct Investments and some environmental aspect such as natural resources, economic freedom and free access.

Please find enclosed my comments.

The abstract is very well written and clear. This is not the case of the Introduction. The reader finally gets into the aim of the paper just at the end of the Section (line 99). The Introduction starts with a paragraph on the importance of FDIs, reporting some evidence on the Commonwealth of Independent States (CIS). This part is really to broad. After the first paragraph, I would point out the aim of the paper and them moving to the literature contribution at line 65 (“Despite ..”). The Introduction has to be improved significantly.

Part 2 on the determinants of FDI: everyone in Academia knows that FDI have been studied quite deeply and intensively. Therefore, there is no need to point it out in the first paragraph (lines 112-117). Move straightforward to the different types of FDIs or just point it out shortly before.

Data description is the right place to move lines 50-63 from the Introduction Section. By commenting on the FDI flows of the CIS Countries in here, motivates why you choose exactly these countries in that period.

The description of the model in Section 3.2 has to be corrected: in the equation you put epsilon as the error term, while in the description it has been changed in u. Moreover, there is no needs to repeat the description of all the variables that you include in the equation, as far as you have already mentioned them in the previous section.

The model is nice and well specified. I have no comments on variable selection and interpretation, neither on the part of the robustness checks.

I will avoid the Discussion section and include the first part in the results section. Then I will move the second part (from line 27 onward) in the conclusion section, which is quite short and do not add major information.

Overall, the study is well conducted, and I really like the empirical part. However, I really push you to highlight the aim pf the paper and the novelty of your results, because in the current way the reader can not appreciate you research.

I hope that with these minor comments you can publish the paper.

Good luck!

Reviewer 2 Report

Hello,

Enjoyed reading your paper. It was a good paper, but in my opinion would benefit from some fine-tuning.

The paper's topic was interesting, though not particularly original. The findings could be found from previous literature. Econometrically the paper was well done. Towards the end, the authors added interpretations that were not in the paper.

REVIEW

Sustainability SUSTAINABILITY-2020-738792

Title: Foreign Direct Investment, Natural Resources, Economic Freedom, and Sea-Access: Evidence from the Commonwealth of Independent States

Summary: The authors aim to explain FDI-flows to 12 CIS-countries. They find natural resources, IEF and Sea-access important.

Impression: A lot of good work has gone into the paper. The findings are not much different from previous literature. Some claims are made which are not supported by empirical findings.

Page, row:

1, 2-5. The title should reflect more clearly the goal of the paper.

1, 17. CIS is not defined.

1, 23. Corruption is a big item in the CIS countries. It is left out of the study.

1, 37. FDI is not defined.

1, 39. Why the big drop in FDI?

1, 41. Since when?

2, 42. How significant?

2, 54-55. According to the Economic Freedom of the World, the 12 CIS countries are far from alike. Georgia is ranked #16, while Turkmenistan is # 177 in the 2019 edition.

2, 52-53. Make relative to GDP.

2, 60-61. Note China’s GDP per capita.

2, 76-77. Insert here the IEF table, including corruption.

3, 112. Extensive is not needed, a short one for sure.

3, 166-117. Syntax.

3, 119-124. The OLI is not well explained.

3, 133-134. What are the crowding-out effects of resource FDI. Please explain.

4, 145. Social – not explained in the manuscript.

4, 147. The political dimension is left out of the manuscript. Check the Freedom in the World index.

4, 150-160. Repetition.

5, 183. The World Bank (2011, note 327) lists the most important factors behind FDI: Market size and growth, EFW, Trade openness, infrastructure, political stability, labor cost and cultural links. The study leaves out market growth, political stability, labor cost and cultural links.

6, 226. Explain revealed comparative advantage.

6, 239. Rule of law is important but it’s nowhere in the regressions (other than its 10% share in the IEF).

6, 244. Ease of doing business is not included in the regressions.

6, 259. There were control variables already before. Additional control variables?

6, 284. There is a lack of a holistic measure for infrastructure.

6, 299. In Table 2, include units.

10, 363. A higher government spending and lower tax burden don’t go together. In the IEF governments get lower points for higher spending.

  1. The only natural resource that is positively correlated with FDI seems to be oil/petroleum.
  2. Access to credit is not part of the descriptive statistics nor any regression. The same applies for non-tariff trade barriers.
  3. 158 and 168%, respectively. Where did these come from?

1, 10. Capital flows was not measured anywhere.

1, 17. Domestic savings was not measured anywhere.

1, 19. Legal systems was not measured anywhere.

1, 22. Business regulations was not measured anywhere (other than as in the above, as a small part of the IEF).

1, 35. Speculation.

2, 43. Some natural resources…

2, 63. That may be true, but some documentation would be in order.

Thanks for the interesting paper.

Reviewer 3 Report

well expressed. worthwhile study. i only have two suggestions. one footnote says that two of the authors contributed equally to the study. how about the other authors? i think that footnote needs to clarify what the contributions of all of the authors are or else be eliminated. Secondly, the abstract says that the size of government is one of the measures of economic freedom. then the text says the economic freedom variable is simply an index. I believe the former is incorrect and the latter is correct.

Round 2

Reviewer 1 Report

Comments to the Authors

I congratulate with the authors for the intensive work they have conduct to improve the paper.

These minor comments concern the Conclusion Section, that in my view is still too short and does not highlight a clear policy implication, or from a practitioner viewpoint if you don’t feel like giving policy advices. The paper has done a great job, but it lacks a soundly conclusion. Could you please improve this part? Maybe you can add also something about how you can contribute with this article to the existing literature.

Finally, there are some typos and minor grammar adjustments (such as the verb tense; analysed at the beginning of the section, while the next sentence is at present – reveal).

Something that you might decide to keep or remove, is the sentence “[7], [32], and [33] are some of the major surveys in the 100 econometric literature on the determinants of FDI.”, it does not add any significant meaning to the paragraph, and it make sense just if you articulate it saying something that you add with respect to these surveys.

Congratulations for your job and I hope the best for your research!
